# Associations between probable REM sleep behavior disorder, olfactory disturbance, and clinical symptoms in Parkinson's disease: A multicenter cross-sectional study

**Mutsumi Iijima**[1]*, **Yasuyuki Okuma**[2], **Keisuke Suzuki**[3], **Fumihito Yoshii**[4], **Shigeru Nogawa**[5], **Takashi Osada**[6,7], **Koichi Hirata**[3], **Kazuo Kitagawa**[1], **Nobutaka Hattori**[8]

1 Department of Neurology, Tokyo Women's Medical University School of Medicine, Tokyo, Japan,
2 Department of Neurology, Juntendo University Shizuoka Hospital, Shizuoka, Japan, 3 Department of Neurology, Dokkyo Medical University, Tochigi, Japan, 4 Department of Neurology, Saiseikai Shonan Hiratsuka Hospital, Hiratsuka, Kanagawa, Japan, 5 Department of Neurology, Tokai University Hachioji Hospital, Tokyo, Japan, 6 Department of Neurology and Cerebrovascular Medicine, Saitama Medical University, International Medical Center, Saitama, Japan, 7 Department of Neurology, Keio University School of Medicine, Tokyo, Japan, 8 Department of Neurology, Juntendo University School of Medicine, Tokyo, Japan

* iijima.mutsumi@twmu.ac.jp

**Data Availability Statement:** All contained within the manuscript.

## Abstract

### Background

Rapid eye movement sleep behavior disorder (RBD) and olfactory dysfunction are useful for early diagnosis of Parkinson's disease (PD). RBD and severe olfactory dysfunction are also regarded as risk factors for cognitive impairment in PD. This study aimed to assess the associations between RBD, olfactory function, and clinical symptoms in patients with PD.

### Methods

The participants were 404 patients with non-demented PD. Probable RBD (pRBD) was determined using the Japanese version of the RBD screening questionnaire (RBDSQ-J) and the RBD Single-Question Screen (RBD1Q). Olfactory function was evaluated using the odor identification test for Japanese. Clinical symptoms were evaluated using the Movement Disorder Society Revision of the Unified PD Rating Scale (MDS-UPDRS) parts I–IV.

### Results

In total, 134 (33.2%) patients indicated a history of pRBD as determined by the RBD1Q and 136 (33.7%) by the RBDSQ-J based on a cutoff value of 6 points. Moreover, 101 patients were diagnosed as pRBD by both questionnaires, 35 by the RBDSQ-J only, and 33 by the RBD1Q only. The MDS-UPDRS parts I–III scores were significantly higher and disease duration significantly longer in the pRBD group. pRBD was significantly associated with male gender and the MDS-UPDRS part I score. The olfactory identification function was significantly reduced in the pRBD group.

**Funding:** The authors report no sources of funding.

**Competing interests:** The authors have declared that no competing interests exist.

## Conclusions

About 33% of the patients with PD had pRBD based on the questionnaires, and both motor and non-motor functions were significantly decreased in these patients. These results suggest that more extensive degeneration occurred in patients with non-demented PD with RBD.

## 1. Introduction

Rapid eye movement sleep behavior disorder (RBD) is a form of parasomnia in which patients develop limb or body movements as a result of dream-enactment behavior [1]. The estimated prevalence of RBD in the general population is only about 0.5%; however, in patients with neurodegenerative disorders, such as Parkinson's disease (PD), multiple system atrophy, and dementia with Lewy bodies (DLB), the prevalence is much higher [2–4]. The prevalence of RBD in patients with PD has been reported as 30–60% [2–14], and 33–45% when diagnosed using polysomnography (PSG) [4, 5, 9]. RBD is one of the early manifestations preceding the onset of typical motor symptoms in patients with PD, including impaired visual and olfactory discrimination and cardiac sympathetic denervation [2, 4, 12, 15, 16–18]. Compared with patients with PD without RBD, those with RBD have also been reported to have more cognitive impairment [4, 18, 19] and greater prevalences of gait freezing, falls, rigidity [2, 4, 19], orthostatic hypotension [4, 9, 20], and visual hallucinations (VHs) [12, 20–22]. The presence of RBD is a predictor for motor progression and cognitive decline in PD with low α-synuclein levels of cerebral spinal fluid [19]. However, motor and cognitive functions in newly diagnosed patients with PD with RBD do not differ from those in patients without RBD [23].

The RBD screening questionnaire (RBDSQ), a 10-item, patient self-rating tool, was proposed for RBD diagnosis as confirmed by PSG [24]. The Japanese version of the RBDSQ (RBDSQ-J) was found to have excellent sensitivity and specificity compared with controls and patients who had obstructive sleep apnea [25]. The RBDSQ-J is thought to be the most appropriate tool for screening for RBD in PD. The RBDSQ-J has a sensitivity of 80% and a specificity of 55% to diagnose RBD with a cutoff of 5 points when compared with standard RBD diagnostic criteria using PSG in PD [25]. When a cutoff of 6 points is used for the RBDSQ-J, the sensitivity and specificity are much higher, at 84.2% and 96.2%, respectively, compared with a cutoff of 5 points [26]. The RBD Single-Question Screen (RBD1Q), a screening questionnaire for dream enactment that utilizes simple yes/no responses, has been reported to have a sensitivity of 93.8% and a specificity of 87.2% [27]. Recently, the RBD1Q showed a sensitivity of 67.7%, a specificity of 82.9%, a positive predictive value (PPV) of 87.5%, and a negative predictive value (NPV) of 59.2% [28].

In this multicenter study, we aimed to investigate the prevalence of probable RBD (pRBD) in Japanese patients with PD and to assess the characteristics of motor and non-motor symptoms in patients with PD with pRBD using the RBDSQ-J with a cutoff of 6 points and the RBD1Q. In addition, we examined whether there was a difference in the results between the RBDSQ-J and the RBD1Q questionnaires.

## 2. Methods

### 2.1 Participants

A total of 423 patients with PD agreed to participate in this study during the registration period from February 2015 to March 2017. A clinical diagnosis of PD was reached in accordance with the UK Brain Bank Criteria [29]. Cognitive function was evaluated using the Mini-Mental State Examination (MMSE). The exclusion criteria were dementia or a score of < 24 on the

MMSE. Eighteen patients were excluded because of a score < 24 on the MMSE and one patient because of providing insufficient information. The participants were 404 patients with non-demented PD (188 men, 216 women; mean age ± standard deviation [SD], 68.7 ± 8.0 years; age range, 39–84 years) who had been seen at the department of neurology in seven hospitals in Japan. The mean ± SD disease duration was 6.5 ± 4.8 years (range, 1–27 years). PD severity (in the patient's "on" state) was graded according to the Hoehn and Yahr (HY) scale; the results showed 49, 245, 89, 16, and 5 patients as grades I–V, respectively. All patients underwent computed magnetic resonance imaging (MRI) to exclude other potential causes of parkinsonism or nasal sinus diseases.

## 2.2 Evaluation of clinical symptoms

Clinical symptoms were evaluated using the Japanese version of the Movement Disorder Society Revision of the Unified PD Rating Scale (MDS-UPDRS) parts I–IV. Motor score (part III) [30] was evaluated during the "on" state. To assess pRBD, we used the RBD1Q [27] and RBDSQ-J [25]. The cutoff for the RBDSQ-J was determined as 6 points based on a previous report [26].

## 2.3 Assessment of olfactory function

The odor identification test for Japanese, either a stick type (Odor Stick Identification Test for Japanese, Daiichi Yakuhin Co., Tokyo, Japan) or a card type (Open Essence, Wako Junyaku Kogyo, Co., Osaka, Japan), was used to assess olfactory function [31]. Both odor identification tests consist of the following same 12 odorants: perfume, rose, condensed milk, Japanese orange, curry, roasted garlic, fermented beans/sweaty socks, cooking gas, menthol, India ink, wood, and Japanese cypress. In this test, the participants choose one of six possible answers from four entities associated with the odors, one of which is correct, and two others (unknown and not detected). All participants were directed to avoid eating and smoking 30 minutes prior to the test.

This study was approved by the Committee of Medical Ethics of Tokyo Women's Medical University (approval No. 3316) and the ethics review committees of each study site, and conducted in compliance with the Ethical Guidelines for Clinical Studies in Japan and the Declaration of Helsinki. Written informed consent was obtained from all patients before the study began.

## 2.4 Statistical analyses

The results are expressed as mean ± SD. JMP Pro statistical software (version 12; SAS Institute, Tokyo, Japan) was used for the statistical analysis. Age, duration of illness, levodopa equivalent dose (LED), and the MDS-UPDRS total and sub-item scores were compared between the pRBD (+) and pRBD (–) groups using the Mann–Whitney U test or Student's *t*–test. The HY scale scores were compared between two groups using Spearman's rank correlation coefficients, and the number of correct answers on the odor identification using Welch's t-test. The chi-squared test was used to compare categorical variables between the pRBD (+) and pRBD (–) groups and the number of patients with pRBD according to the RBDSQ-J and RBD1Q. The agreement between the RBDSQ-J and RBD1Q was evaluated by Cohen's Kappa coefficient. Next, a logistic regression analysis was performed using pRBD as the dependent variable and age, sex, disease duration, MDS-UPDRS parts I–IV total scores, and odor identification scores as the independent variables. Furthermore, sub-items of each part of the MDS-UPDRS that contributed to pRBD were examined using logistic analysis. Multiple regression analysis using age, sex, disease duration, MDS-UPDRS parts I–IV total scores, and number of correct

answers on the odor identification test as independent variables was then performed to predict RBDSQ-J scores. *P* values < 0.05 were regarded as significant.

## 3. Results

### 3.1 Patients' characteristics

Table 1 shows the total scores on the MDS-UPDRS parts I–IV. MMSE scores ranged from 24 to 30 points (mean ± SD, 28.6 ± 1.8 points). In total, 344 patients were taking anti-parkinsonian medication, and the LED was 390 ± 227 mg/day. In addition, 60 patients were de novo. The mean number of correct scores on the odor identification test was 4.8 ± 2.7, and 197 patients (48.7%) had severe hyposmia (4 points or less). When the cutoff of RBDSQ-J was 6 points, 136 patients (33.7%) were considered pRBD (+) and 268 (66.3%) pRBD (–). On the RBD1Q, 134 patients (33.2%) answered "yes" and 270 (66.8%) answered "no". The number of patients with pRBD did not significantly differ based on the RBDSQ-J and RBD1Q. The two questionnaires showed moderate agreement, with a Cohen's kappa coefficient of 0.62. In addition, 101 patients (25.0%) were diagnosed as pRBD by both questionnaires, 35 (8.7%) by the RBDSQ-J only (pRBDSQ-J group), and 33 (8.4%) by the RBD1Q only (pRBD1Q group). A comparison of patients with pRBD between the pRBDSQ-J and pRBD1Q groups showed no significant difference in age, disease duration, MMSE, total scores on the MDS-UPDRS parts II, III, and IV, and olfactory function; however, significant differences were seen in sex, HY scale, and total score of the MDS-UPDRS part I (13.6 points in the pRBDSQ-J group vs. 9.6 points in the pRBD1Q group, p < 0.05, Table 2). Sleep (p = 0.04) and fatigue scores (p = 0.006) in the sub-items of the MDS-UPDRS part I were significantly higher in patients in the pRBDSQ-J group.

### 3.2 Comparison of parameters between PD with and without pRBD

MDS-UPDRS parts I–III scores and the LED were significantly higher (p < 0.001) and disease duration was significantly longer (p < 0.05) in the pRBD (+) than in the pRBD (–) group

**Table 1. Clinical characteristics of total patients with Parkinson's disease.**

| | |
|---|---|
| Number of patients | 404 |
| Gender (male/female) | 188/216 |
| Age (years) | 68.7 ± 8.0 (39–84) |
| Disease duration (years) | 6.5 ± 4.8 (1–27) |
| Mini-Mental State Examination | 28.6 ± 1.8 (24–30) |
| Hoehn and Yahr stage (on phase) | 2.3 ± 0.7 (1–5) |
| Levodopa-equivalent daily dose (mg/day) | 390 ± 227 (0–1500) |
| MDS UPDRS part I | 10.0 ± 6.3 (1–41) |
| part II | 12.4 ± 8.6 (0–42) |
| part III | 45.3 ± 22.4 (2–112) |
| part IV | 1.9 ± 3.3 (1–21) |
| RBDSQ-J score | 4.5 ± 2.8 (1–13) |
| Patient numbers of pRBD | RBDSQ-J: 136 (33.7%) |
| | RBD1Q: 134 (33.2%) |
| Correct answer of odor identification test | 4.8 ± 2.7 (0–12) |

Values are mean ± SD (range).

MDS-UPDRS: Movement Disorder Society Revision of the Unified Parkinson's disease rating scale, RBDSQ-J: REM sleep behavior disorder screening questionnaire-Japanese version, RBD1Q: the RBD Single-Question Screen, pRBD: probable REM sleep behavior disorder.

**Table 2. Comparison of characteristics between patients with PD and pRBD only by the RBDSQ-J and RBD1Q.**

|  | RBDSQ-J | RBD1Q | p value |
|---|---|---|---|
| N | 35 | 33 |  |
| Age (year) | 68.6 ± 7.9 | 65.4 ±8.6 | 0.11 |
| SEX (n) M:F | 15:20 | 20:13 | 0.023 |
| Duration (year) | 6.5 ± 5.1 | 5.7 ± 4.2 | 0.475 |
| Hoehn & Yahr scale (on period) | 2.5 ± 0.9 | 2.1 ± 0.6 | 0.048 |
| Mini-Mental State Examination | 28.2 ± 1.7 | 28.5 ±1.9 | 0.60 |
| LED (mg/day) | 538 ± 603 (n:30) | 373 ± 225 (n:30) | 0.164 |
| MDS-UPDRS UPDRS I | 13.0 ± 8.1 | 9.6 ± 5.2 | 0.045 |
| UPDRS II | 15.9 ± 10.4 | 12.1 ± 7.0 | 0.081 |
| UPDRS III | 51.2 ± 20.0 | 42.1 ± 24.6 | 0.096 |
| UPDRS IV | 2.2± 3.7 | 1.2± 3.4 | 0.24 |
| Correct answer of odor identification test | 4.3 ± 2.1 | 3.7 ± 2.1 | 0.31 |

Values are mean ± SD (range), RBDSQ-J: REM sleep behavior disorder screening questionnaire-Japanese version, RBD1Q: the RBD Single-Question Screen, pRBD: probable REM sleep behavior disorder, LED: levodopa equivalent dose, MDS-UPDRS: Movement Disorder Society Revision of the Unified Parkinson's disease rating scale.

based on the RBDSQ-J (Table 3). These results were the same as those based on the RBD1Q, excluding disease duration, the HY scale, and the MMSE. The mean numbers of correct answers on the olfactory identification test were 5.0 ± 2.9 in the pRBD (−) group and 4.4 ± 2.4 in the pRBD (+) group based on the RBDSQ-J, and 5.0 ± 2.8 in the pRBD (−) group and 4.3 ± 2.5 in the pRBD (+) group based on the RBD1Q, which was significantly lower (p = 0.01) in the pRBD (+) than in the pRBD (−) group (Table 3). Logistic regression analysis with pRBD diagnosed by RBD-SQJ as the objective variable showed that male sex (male = 1, female = 2, odds ratio [OR]: 1.69, 95% confidence interval [CI]: 1.08–2.65, p = 0.021) and MDS-UPDRS part I total score (OR: 1.05, 95% CI: 1.01–1.09, p = 0.026) were contributing factors to pRBD.

### 3.3 Comparison of sub-items of the MDS-UPDRS parts I–III between PD with and without pRBD based on the RBDSQ-J

The mean sub-item scores on the MDS-UPDRS parts I–III are shown in Tables 4 and 5. Many sub-items were significantly higher in the pRBD (+) than in the RBD (−) group. As a result of examining the involved of sub-items of pRBD for each part of the MDS-UPDRS by logistic regression analysis, VHs (p = 0.047) and apathy (p = 0.030) in part I, dressing (p = 0.032) in part II, and gait (p = 0.010) and pronation-supination movements (p = 0.018) in part III were related clinical features of pRBD in patients with PD (Table 6).

Multiple regression analysis performed to predict RBDSQ-J scores showed that sex (p = 0.033) and total score on the MDS-UPDRS part I (p = 0.001) were significant predictors of pRBD in patients with PD. In the sub-items of the MDS-UPDRS part I related to RBDSQ-J scores, VH (p = 0.018), sleep problems (p < 0.001), and daytime sleepiness (p = 0.028) were significantly associated with RBDSQ-J score.

## 4. Discussion

This multicenter study attempted to clarify the prevalence of pRBD using RBD questionnaires conducted on patients with PD, as well as the relationship between pRBD and motor and non-motor symptoms. Male sex and the MDS-UPDRS part I total score were found to be contributing factors to pRBD. The prevalence of pRBD was 33% when the cutoff value was 6 for the

**Table 3. Comparison between patients with PD with and without pRBD.**

|  | pRBD (-) | pRBD (+) | p value |
|---|---|---|---|
| Evaluated by RBDSQ-J |  |  |  |
| N | 268 | 136 |  |
| Age (year) | 67.4 ± 8.0 | 68.3 ±8.1 | 0.27 |
| SEX (n) M:F | 111:157 | 77:59 | < 0.005 |
| Duration (year) | 6.1 ± 4.3 | 7.2 ± 5.5 | 0.03 |
| Hoehn & Yahr scale | 2.2 ± 0.7 | 2.5 ± 0.8 | < 0.001 |
| Mini-Mental State Examination | 28.7 ± 1.7 | 28.3 ±1.8 | 0.06 |
| LED (mg/day) | 426 ± 349 | 573 ± 461 | < 0.001 |
| MDS-UPDRS UPDRS I | 9.0 ± 6.1 | 11.9 ± 6.3 | < 0.0001 |
| UPDRS II | 10.9 ± 7.9 | 15.2 ± 9.2 | < 0.0001 |
| UPDRS III | 42.5 ± 22.2 | 50.7 ± 21.8 | 0.0004 |
| UPDRS IV | 1.7 ± 3.2 | 2.2 ± 3.4 | 0.16 |
| Correct answer of odor identification test | 5.0 ± 2.9 | 4.4 ± 2.4 | < 0.05 |
| Evaluated by RBD 1Q |  |  |  |
| N | 270 | 134 |  |
| Age (year) | 67.8 ± 7.8 | 67.5 ± 8.4 | 0.73 |
| SEX (n) M:F | 106:164 | 82: 52 | < 0.0001 |
| Duration (year) | 6.2 ± 4.5 | 7.0 ± 5.3 | 0.11 |
| Hoehn & Yahr scale | 2.2 ± 0.7 | 2.4 ± 0.8 | 0.13 |
| Mini-Mental State Examination | 28.6 ± 1.7 | 28.4 ± 1.9 | 0.14 |
| LED (mg/day) | 447 ± 401 | 533 ± 384 | < 0.05 |
| MDS-UPDRS UPDRS I | 9.4 ± 6.6 | 11.1 ± 5.5 | 0.013 |
| UPDRS II | 11.4 ± 8.5 | 14.3 ± 8.4 | 0.002 |
| UPDRS III | 43.7 ± 21.8 | 48.5 ± 23.2 | 0.042 |
| UPDRS IV | 1.8 ± 3.3 | 2.0 ± 3.2 | 0.73 |
| Correct answer of odor identification test | 5.0 ± 2.8 | 4.3 ± 2.5 | 0.01 |

Values are mean ± SD (range), RBDSQ-J: REM sleep behavior disorder screening questionnaire-Japanese version, RBD1Q: the RBD Single-Question Screen, pRBD: probable REM sleep behavior disorder, LED: levodopa equivalent dose, MDS-UPDRS: Movement Disorder Society Revision of the Unified Parkinson's disease rating scale.

RBDSQ-J, and was similar for the RBD1Q. In the previous reports using a cutoff value of 6 for the RBDSQ, the prevalence of pRBD ranged from 17% to 56% [10, 13, 20, 22, 26, 28]. The sensitivity and specificity of the RBDSQ for the PD cohort were 0.44–0.84 and 0.55–0.93, respectively, in previous reports using a cutoff score of 6 for patients with PD [15, 26, 28, 32, 33]. Factors that have been reported to affect RBD in PD include cognitive decline and gender [4]. In this study, patients with marked dementia were excluded; therefore, the difference in the prevalence of RBD in patients with PD may have been influenced by differences in patient backgrounds.

This study found moderate agreement between the RBDSQ-J and RBD1Q, but 35 patients were diagnosed as pRBD by the RBDSQ-J only and 33 by the RBD1Q only. The HY scale, total score on the MDS-UPDRS I, and sleep and fatigue sub-items scores on the MDS-UPDRS I were higher in patients with pRBD diagnosed by the RBDSQ-J only. This result suggests that severe motor dysfunction, sleep problems, and fatigue affected the diagnosis of pRBD by the RBDSQ. The difference in the ratio of gender differences between the two groups is not clear. Since it has been confirmed that the proportion of males is high among patients with RBD, pRBD evaluation by the RBD1Q can be considered to reflect gender differences more accurately.

**Table 4. Comparison of sub-items on the MDS-UPDRS parts I and II between patients with PD with and without pRBD as evaluated by the Japanese version of the RBD screening questionnaire.**

|  | pRBD (-) | pRBD (+) | p value |
|---|---|---|---|
| Part I items |  |  |  |
| Cognitive impairment | 0.20 | 0.24 | 0.48 |
| Hallucinations | 0.12 | 0.31 | < 0.0001 |
| Depressed mood | 0.38 | 0.45 | 0.34 |
| Anxious mood | 0.53 | 0.60 | 0.43 |
| Apathy | 0.31 | 0.51 | 0.007 |
| Dopamine dysregulation syndrome | 0.03 | 0.10 | 0.021 |
| Sleep problems | 0.95 | 1.41 | < 0.0001 |
| Daytime sleepiness | 1.25 | 1.56 | 0.0005 |
| Pain | 0.80 | 1.01 | 0.055 |
| Urinary problems | 0.95 | 1.40 | < 0.0001 |
| Constipation | 1.30 | 1.70 | 0.002 |
| Light headedness on standing | 0.42 | 0.51 | 0.239 |
| Fatigue | 0.98 | 1.28 | 0.003 |
| Part II items |  |  |  |
| Speech | 0.75 | 1.13 | 0.0001 |
| Saliva & Drooling | 0.92 | 1.34 | 0.0006 |
| Chewing & Swallowing | 0.38 | 0.51 | 0.086 |
| Eating tasks | 0.68 | 0.86 | 0.025 |
| Dressing | 0.87 | 1.23 | < 0.0001 |
| Hygiene | 0.75 | 1.00 | 0.002 |
| Handwriting | 0.88 | 1.15 | 0.004 |
| Doing hobbies | 1.01 | 1.40 | 0.0005 |
| Turning in bed | 0.75 | 1.14 | < 0.0001 |
| Tremor | 1.00 | 1.24 | 0.005 |
| Getting out of bed, or deep chair | 0.96 | 1.41 | < 0.0001 |
| Walking and balance | 1.20 | 1.61 | 0.0002 |
| Freezing | 0.81 | 1.21 | 0.001 |

pRBD: probable REM sleep behavior disorder.

 Total scores on MDS-UPDRS parts I–III and the LED were significantly higher in the pRBD (+) than in the pRBD (−) group. Previous studies have reported that scores on the MDS-UPDRS parts II–IV were higher in PD with pRBD than without RBD [11], and drug-naïve patients with early PD with RBD had higher scores on MDS-UPDRS parts I and II than did patients with PD without RBD [19]. Among the non-motor symptoms on MDS-UPDRS part I, VHs and apathy were related to pRBD in this study. An association between VHs and RBD in patients with PD has been reported in numerous studies [11, 19, 20, 22]. Liu et al. [20] compared the clinical symptoms of PD with and without RBD with RBDSQ cutoff values of 5, 6, and 7 points. A significant difference in VHs was observed between the two groups at a cut-off of 6 points and above, but not at a cutoff value of 5 points. In addition, the present study revealed that VHs, sleep problems, and daytime sleepiness on MDS-UPDRS part I affected the RBDSQ-J scores. VHs, male gender, and MDS-UPDRS part II scores have been reported to be contributing factors to pRBD in patients with PD [11]. Although there are few reports about the relationship between apathy and RBD, more severe apathy symptoms were recently reported in patients with PD with than without RBD, independent of age and years [34].

**Table 5. Comparison of sub-items on the MDS-UPDRS part III between patients with PD with and without pRBD as evaluated by the Japanese version of the RBD screening questionnaire.**

|  | RBD (-) | RBD (+) | P value |
|---|---|---|---|
| Part III items |  |  |  |
| Speech | 0.75 | 1.04 | 0.0005 |
| Facial expression | 0.81 | 1.01 | 0.013 |
| Rigidity of neck and four extremities | 5.84 | 6.93 | 0.001 |
| Finger taps | 2.15 | 2.56 | 0.037 |
| Hand movements | 1.15 | 1.46 | 0.027 |
| Pronation/supination | 2.50 | 3.04 | 0.001 |
| Toe tapping | 2.18 | 2.51 | 0.043 |
| Leg agility | 1.63 | 1.89 | 0.06 |
| Arising from chair | 0.46 | 0.85 | < 0.0001 |
| Gait | 0.98 | 1.49 | < 0.0001 |
| Freezing of gait | 0.26 | 0.52 | 0.003 |
| Postural stability | 0.72 | 1.11 | 0.002 |
| Posture | 0.96 | 1.41 | < 0.0001 |
| Global spontaneity of movement | 1.04 | 1.46 | < 0.0001 |
| Postural tremor of hands | 0.80 | 0.75 | 0.60 |
| Kinetic tremor of hands | 0.76 | 0.77 | 0.90 |
| Rest tremor amplitude | 0.86 | 0.87 | 0.97 |
| Constancy of rest tremor | 0.63 | 0.57 | 0.60 |

pRBD: probable REM sleep behavior disorder.

Apathy is strongly related to the limbic system, and evidence for the involvement of limbic structures in the pathophysiology of RBD is increasing; therefore, lesions associated with apathy and RBD were suggested in regard to the limbic system [34].

Motor functions as assessed by the MDS-UPDRS part III were diminished in the PD with pRBD (+) group. Among the sub-items on MDS-UPDRS part III, gait and pronation–supination movements were associated with pRBD. Previous reports have revealed that falls and gait freezing were more frequent in PD with than without RBD [4].

The mechanisms underlying the pathogenesis of RBD remain unclear. In animal models, rapid eye movement sleep regulation involves brain stem centers such as the glutamatergic subcoeruleus/sublateral dorsal nucleus, noradrenergic locus coeruleus, cholinergic

**Table 6. Logistic regression analysis of pRBD as evaluated by the Japanese version of the RBD screening questionnaire based on the sub-items of each part of the MDS-UPDRS.**

|  | OR | 95% CI | p value |
|---|---|---|---|
| MDS-UPDRS part I |  |  |  |
| Hallucinations | 1.70 | 1.01–2.88 | 0.047 |
| Apathy | 1.59 | 1.05–2.45 | 0.030 |
| MDS-UPDRS part II |  |  |  |
| Dressing | 1.61 | 1.05–2.45 | 0.032 |
| MDS-UPDRS part III |  |  |  |
| Gait | 1.84 | 1.17–2.96 | 0.010 |
| Pronation-supination movements | 1.29 | 1.04–1.60 | 0.018 |

pRBD: probable REM sleep behavior disorder.

pedunculopontine (PPN), and laterodorsal tegmental nuclei, as well as the medullary magno-cellular reticular formation, with additional modulation by the hypothalamus, thalamus, substantia nigra, basal forebrain, limbic system, and frontal cortex [4]. Patients with PD and pRBD showed smaller volumes than patients without RBD and healthy controls in the ponto-mesencephalic tegmentum, medullary reticular formation, hypothalamus, thalamus, putamen, amygdala, and anterior cingulate cortex [35]. In addition, MRI of cortical volume was particularly thin in the right perisylvian area and inferior temporal lobe in PD with RBD [36]. Autopsy studies have found that cholinergic cell density in the PPN is significantly lower in patients with PD with VHs compared with those with DLB with VHs [36], and that PPN hypofunction is associated with VHs [37]. A loss of cholinergic neurons in the PPN has been reported to be negatively correlated with the HY scale [38]. The association between pRBD with VHs and gait dysfunction in patients with PD in this study suggests the dysfunction of the PPN and cerebral lesions projecting from the PPN.

Regarding the relationship between RBD and olfactory dysfunction, scores on the olfactory identification function test were significantly lower in patients with PD and pRBD as diagnosed in this study. A previous report showed that scores on the University of Pennsylvania Smell Identification Test were lower in patients with than without RBD [19]. Lewy bodies first develop in the olfactory bulb and anterior olfactory nucleus (Braak stage I), and then progress in the amygdala, the hippocampus, the piriform cortex, and the entorhinal and the orbitofrontal cortices, which constitute the mesolimbic pathway [39]. The mesolimbic pathway is also important in the pathogenesis of RBD [2]. Severe olfactory dysfunction in patients with PD and pRBD suggests that the mesolimbic pathway, which has been considered a common lesion of RBD and olfactory dysfunction, was more impaired in such patients compared with those without pRBD. RBD and olfactory dysfunction are thought to be predictors of PD and to be associated with cognitive impairment [19, 40]. However, among patients with PD with RBD, only low Aβ42 and α-synuclein levels have been reported to be predictors of cognitive decline [19]. Our results may have been influenced by the exclusion of patients with an obvious cognitive impairment.

This study had some limitations. First, RBD diagnosis was not certain because it was done by a questionnaire rather than by PSG. Second, the relationship between RBD and cognitive function may not have been able to be evaluated because patients with obvious cognitive decline were excluded.

## 5. Conclusion

pRBD as diagnosed using the RBDSQ-J and RBD1Q questionnaires was found in approximately 33% of the patients with PD without obvious dementia. Motor and non-motor functions as assessed by the MDS-UPDRS were significantly lower in patients with PD and pRBD. These results suggest that more extensive degeneration occurs in patients with PD and pRBD.

## Author Contributions

**Conceptualization:** Mutsumi Iijima, Yasuyuki Okuma, Keisuke Suzuki, Fumihito Yoshii, Shi-geru Nogawa, Takashi Osada, Koichi Hirata, Kazuo Kitagawa, Nobutaka Hattori.

**Data curation:** Mutsumi Iijima, Yasuyuki Okuma, Keisuke Suzuki, Fumihito Yoshii, Shigeru Nogawa, Takashi Osada, Koichi Hirata.

**Formal analysis:** Mutsumi Iijima.

**Investigation:** Mutsumi Iijima, Yasuyuki Okuma, Keisuke Suzuki, Fumihito Yoshii, Kazuo Kitagawa.

**Methodology:** Mutsumi Iijima, Yasuyuki Okuma.

**Project administration:** Nobutaka Hattori.

**Supervision:** Nobutaka Hattori.

**Writing – original draft:** Mutsumi Iijima.

**Writing – review & editing:** Yasuyuki Okuma, Keisuke Suzuki, Fumihito Yoshii, Shigeru Nogawa, Takashi Osada, Koichi Hirata, Kazuo Kitagawa, Nobutaka Hattori.

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
