## [Decision Letter · Decision Letter 0]

11 Dec 2020

PONE-D-20-34674

Associations between rapid eye movement sleep behavior disorder, olfactory disturbance, and clinical symptoms in Parkinson’s disease: a multicenter cross-sectional study

PLOS ONE

Dear Dr. Iijima,

Thank you for submitting your manuscript to PLOS ONE. After careful consideration, we feel that it has merit but does not fully meet PLOS ONE’s publication criteria as it currently stands. Therefore, we invite you to submit a revised version of the manuscript that addresses the points raised during the review process.

We look forward to receiving your revised manuscript.

Kind regards,

Mathias Toft, MD, PhD

Academic Editor

PLOS ONE

Journal Requirements:

" The funders had no role in study design, data collection and analysis, decision to publish, or preparation of the manuscript."

Reviewers' comments:

Reviewer's Responses to Questions

**Comments to the Author**

1. Is the manuscript technically sound, and do the data support the conclusions?

Reviewer #1: Yes

Reviewer #2: Yes

2. Has the statistical analysis been performed appropriately and rigorously? 

Reviewer #1: Yes

Reviewer #2: Yes

3. Have the authors made all data underlying the findings in their manuscript fully available?

Reviewer #1: Yes

Reviewer #2: Yes

4. Is the manuscript presented in an intelligible fashion and written in standard English?

Reviewer #1: No

Reviewer #2: Yes

5. Review Comments to the Author

Reviewer #1: The present study is an interesting multicenter study including a significant number of patients. The results are in line with previous reports, and supports earlier findings of more extensive degeneration in PD patients with RBD.

I have some comments:

Page 11, line 167-169: The authors claim tha disease duration was significantly longer in PRBD both on RDBDSQ-J and RBDQ1 ("these results were the same as those based on the RBD1Q.") This does not match the results in table 2, p-values 0.03 and 0.11, respectively.

Page 12, results section: Line 180-186: Describe the statistical method (logistic regression)

Page 13, Discussion: Line 205-207: The sentence is repeated from the abstract and the results section, but not discussed. Do you have any commets on the difference? Are there any particular items on the RBDSQ that apply on those who are negative on RBD1Q? For instance those items that concerns motor activity t night

In general, the language is clear and easily understandable. Some parts, particularly in the discussion, are however somewhat difficult to understand. Furthermore, several sentences should be rephrased.

Some suggestions:

Page 3:Abstract, Methods, line 43: The participants were 404 non-demented PD patients

Page 7, Methods, line 99-100: Eighteen patients were excluded due to a score <24 on the MMSE and one patients was excluded due to insufficient information.

Page 7, line 103: “multicenter hospitals in Japan” – needs to be rephrased. How many hospitals?

Page 8, line 117: Both odor identification tests… (not the both)

Page 13:, line 205: This is the first study to describe..

Page 16, line 254: sentence should be rephrased

Page 17, line 268-269: obvious cognitive decline, not cognitive function

Reviewer #2: The authors aimed to assess the association between RBD, olfactory function and clinical symptoms in patients with Parkinson’s disease.

The manuscript is well written, and the topic is definitely current and quite interesting.

There are few minor points which need to be clarified.

1. The authors should retitle the manuscript using "probable RBD" as vPSG is mandatory for the definite diagnosis of RBD and the authors used screening questionnaires instead.

2. The authors used two widely used screening questionnaire to investigate RBD in their population, namely the Japanese version of RBDSQ and the RBD1Q. The latter has been validated in a cohort of isolated RBD, finding 84.2% of sensitivity and 96.2% of specificity. Recently, RBD1Q has been used in a large cohort of 97 non-demented PD patients, showing 67.7% of sensitivity, 82.9% of specificity, with 87.5% of PPV and 59.2% of NPV when administered prior to any kind of clinical interview. (Figorilli et al. doi: 10.1093/sleep/zsz323).

3. How do the authors explain the differences between the two questionnaires in terms of associations with motor and non-motor symptoms?

6. PLOS authors have the option to publish the peer review history of their article (what does this mean?). If published, this will include your full peer review and any attached files.

Reviewer #1: No

Reviewer #2: No

---

## [Author Response · Author response to Decision Letter 0]

25 Dec 2020

We have responded to reviewer comments in attached file "Response to Reviewers".

---

## [Decision Letter · Decision Letter 1]

14 Jan 2021

PONE-D-20-34674R1

Associations between probable REM sleep behavior disorder, olfactory disturbance, and clinical symptoms in Parkinson’s disease: a multicenter cross-sectional study

PLOS ONE

Dear Dr. Iijima,

Thank you for submitting your manuscript to PLOS ONE. After careful consideration, we feel that it has merit but does not fully meet PLOS ONE’s publication criteria as it currently stands. Therefore, we invite you to submit a revised version of the manuscript that addresses the points raised during the review process.

Please see the remaining comments from one of the reviewers. In addition, further editing of the English language is necessary before the manuscript can be accepted for publication. 

We look forward to receiving your revised manuscript.

Kind regards,

Mathias Toft, MD, PhD

Academic Editor

PLOS ONE

Reviewers' comments:

Reviewer's Responses to Questions

**Comments to the Author**

1. If the authors have adequately addressed your comments raised in a previous round of review and you feel that this manuscript is now acceptable for publication, you may indicate that here to bypass the “Comments to the Author” section, enter your conflict of interest statement in the “Confidential to Editor” section, and submit your "Accept" recommendation.

Reviewer #1: (No Response)

Reviewer #2: All comments have been addressed

2. Is the manuscript technically sound, and do the data support the conclusions?

Reviewer #1: Yes

Reviewer #2: Yes

3. Has the statistical analysis been performed appropriately and rigorously? 

Reviewer #1: Yes

Reviewer #2: Yes

4. Have the authors made all data underlying the findings in their manuscript fully available?

Reviewer #1: Yes

Reviewer #2: Yes

5. Is the manuscript presented in an intelligible fashion and written in standard English?

Reviewer #1: No

Reviewer #2: No

6. Review Comments to the Author

Reviewer #1: The revised manuscript is improved. However, I still have some comments.

1: The term probable RBD should be introduced earlier in the manuscript, not in the statistics section. I would suggest that you introduce the term pRBD in line 89, because you aim to investigate the prevalence of pRBD, not RBD.

2: The term pRBD should be used consistently throughout the manuscript when you refer to your present study, for instance in line 113

3: Line 160: Sentence should be rephrased

4: Line 210: the study showed good agreement (not the good agreement)

5: Line 212 and table 2: Significantly more men diagnosed by RBDSQ-J. Any comments?

6: Line 214/215: Sentence is not easily understandable and should be rephrased.

7: Line 224: As a result, a significant difference…. Should be rephrased, “A significant difference..”

Reviewer #2: The authors adequately addressed all the points raised by the reviewer. However, the English should be revised.

7. PLOS authors have the option to publish the peer review history of their article (what does this mean?). If published, this will include your full peer review and any attached files.

Reviewer #1: No

Reviewer #2: No

---

## [Author Response · Author response to Decision Letter 1]

25 Jan 2021

We have uploaded comments to editors and reviewers to files respectively.

---

## [Decision Letter · Decision Letter 2]

8 Feb 2021

Associations between probable REM sleep behavior disorder, olfactory disturbance, and clinical symptoms in Parkinson’s disease: a multicenter cross-sectional study

PONE-D-20-34674R2

Dear Dr. Iijima,

We’re pleased to inform you that your manuscript has been judged scientifically suitable for publication and will be formally accepted for publication once it meets all outstanding technical requirements.

Kind regards,

Mathias Toft, MD, PhD

Academic Editor

PLOS ONE

Additional Editor Comments (optional):

Reviewers' comments:

Reviewer's Responses to Questions

**Comments to the Author**

1. If the authors have adequately addressed your comments raised in a previous round of review and you feel that this manuscript is now acceptable for publication, you may indicate that here to bypass the “Comments to the Author” section, enter your conflict of interest statement in the “Confidential to Editor” section, and submit your "Accept" recommendation.

Reviewer #1: All comments have been addressed

Reviewer #2: All comments have been addressed

2. Is the manuscript technically sound, and do the data support the conclusions?

Reviewer #1: (No Response)

Reviewer #2: Yes

3. Has the statistical analysis been performed appropriately and rigorously? 

Reviewer #1: (No Response)

Reviewer #2: Yes

4. Have the authors made all data underlying the findings in their manuscript fully available?

Reviewer #1: (No Response)

Reviewer #2: Yes

5. Is the manuscript presented in an intelligible fashion and written in standard English?

Reviewer #1: (No Response)

Reviewer #2: Yes

6. Review Comments to the Author

Reviewer #1: (No Response)

Reviewer #2: (No Response)

7. PLOS authors have the option to publish the peer review history of their article (what does this mean?). If published, this will include your full peer review and any attached files.

Reviewer #1: No

Reviewer #2: No

---

## [Editor Report · Acceptance letter]

10 Feb 2021

PONE-D-20-34674R2 

Associations between probable REM sleep behavior disorder, olfactory disturbance, and clinical symptoms in Parkinson’s disease: a multicenter cross-sectional study 

Dear Dr. Iijima:

I'm pleased to inform you that your manuscript has been deemed suitable for publication in PLOS ONE. Congratulations! Your manuscript is now with our production department. 

Kind regards, 

on behalf of

Dr Mathias Toft 

Academic Editor

PLOS ONE